# VAULT: Vigilant Adversarial Updates via LLM-Driven Retrieval-Augmented Generation for NLI

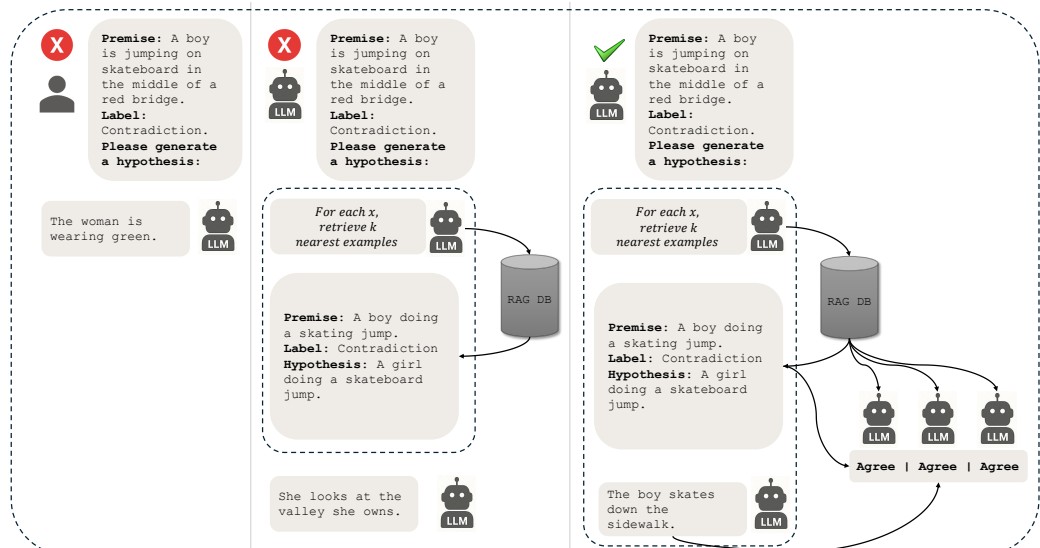

Figure 1: VAULT's stages: on the **left**, direct LLM hypothesis generation (no retrieval or validation); in the **middle**, context retrieval and adversarial hypothesis generation (no validation); and on the **right**, the full pipeline with retrieval, generation, and automated validation before reinjection.

## ABSTRACT

We introduce **VAULT**, a fully automated adversarial RAG pipeline that systematically uncovers and remedies weaknesses in NLI models through three stages: **retrieval**, **adversarial generation**, and **iterative retraining**. First, we perform balanced few-shot retrieval by embedding premises with both semantic (BGE) and lexical (BM25) similarity. Next, we assemble these contexts into LLM prompts to **generate adversarial hypotheses**, which are then validated by an LLM ensemble for label fidelity. Finally, the validated adversarial examples are **injected back** into the training set at increasing mixing ratios, progressively fortifying a zero-shot target NLI model. On standard benchmarks, VAULT elevates RoBERTa-base accuracy from **88.48%** to **92.60%** on SNLI (+4.12%), from **75.04%** to **80.95%** on ANLI (+5.91%), and from **54.67%** to **71.99%** on MultiNLI (+17.32%). It also consistently outperforms prior in-context adversarial methods by up to **2.0%** across datasets. By automating high-quality adversarial data curation at scale, VAULT enables rapid, human-independent robustness improvements in NLI inference tasks.

# 1 INTRODUCTION

Natural language inference (NLI)-the task of determining whether a hypothesis is entailed by, contradicted by, or neutral with respect to a given premise-is fundamental to many downstream NLP applications such as question answering, summarization, and dialogue systems. Despite rapid progress, even state-of-the-art models remain brittle when faced with adversarial or out-of-domain examples, often exploiting spurious lexical cues or failing on simple syntactic variations (Glockner et al., 2018; Carmona et al., 2018). Benchmarks like ANLI (Nie et al., 2019) and manually curated corpora such as SNLI (Bowman et al., 2015) and MultiNLI (Williams et al., 2018) have driven robustness improvements but incur high annotation costs and still leave many failure modes uncovered. More recently, large synthetic datasets like GNLI (Hosseini et al., 2024) have been generated at scale, but their untargeted nature often dilutes the most critical adversarial patterns.

Inspired by these limitations, we introduce VAULT, a fully automated adversarial Retrieval-Augmented Generation (RAG) pipeline that systematically mines and repairs the weak spots of NLI models without any manual labeling. VAULT begins by retrieving balanced few-shot contexts from SNLI using both semantic embeddings (BGE M3 (Chen et al., 2024)) and lexical matching (BM25 (Robertson & Zaragoza, 2009)), then prompts a LLM to generate challenging hypotheses tailored to the model's current weaknesses. Each candidate pair of premise and hypothesis is vetted by an ensemble of three LLMs and only unanimously agreed-upon examples are used for future training of the target model. By iterating this retrieve-generate-validate loop for multiple rounds, VAULT progressively hardens the target NLI model against its own blind spots, focusing data where it matters most.

In a strict zero-shot evaluation on SNLI, ANLI, and MultiNLI test sets, VAULT achieves substantial gains over the original RoBERTa-base accuracy: from 88.48% to 92.13% on SNLI (+3.65%), from 75.04% to 80.27% on ANLI (+5.23%), and from 54.67% to 71.12% on MultiNLI (+16.45%). These improvements exceed those of prior in-context adversarial approaches by at least 2% on each benchmark, demonstrating that fully automated adversarial augmentation can match or surpass human-curated data with only a fraction of the examples.

The key stages of VAULT are:

- **Retrieval:** Embed all SNLI data with a semantic text embedder and retrieve balanced few-shot examples via both semantic and lexical similarity;

- **Generation:** Using the retrieved exmaples, assemble a context to serve as prompt for the LLM and generate challenging hypotheses;

- **Adversarial Filtering:** Pass the generated examples through the target model, and keep only the examples that failed it.

- **Validation & training:** Further filter the generated examples for unanimous agreement among the LLM judges to ensure data correctness. Then use high-confidence examples for training;

- **Iterative Retraining:** repeat all previous steps for multiple rounds to continually strengthen the model.

Our contributions are:

1. An end-to-end automated adversarial RAG pipeline that requires no human annotation and adapts dynamically to a model's weaknesses.

2. A demonstration that targeted synthesis and validation solely via LLMs can yield significant zero-shot and few-shot accuracy gains on multiple NLI benchmarks using an order of magnitude less data than prior synthetic corpora.

3. Empirical evidence that VAULT not only outperforms existing adversarial augmentation methods in effectiveness but also offers superior data efficiency, highlighting a new direction for high-impact robustness improvements.

## 2 BACKGROUND AND RELATED WORK

Improving the robustness and performance of NLI models remains a significant challenge in natural language understanding (Glockner et al., 2018; Carmona et al., 2018). While traditional approaches heavily relied on manually created datasets, such as the Stanford NLI (SNLI) corpus (Bowman et al., 2015), this labor-intensive process highlighted the need for more efficient alternatives. The Multi-Genre NLI (MultiNLI) dataset (Williams et al., 2018) expanded coverage to diverse text genres, and ANLI (Nie et al., 2019) introduced a human-and-model-in-the-loop protocol to collect hard cases, yet all still require extensive annotation effort. More recently, (Kazoom et al., 2025) proposed a training-free adversarial detection framework that leverages retrieval-augmented generation to automatically generate and filter challenging examples without manual labeling. Recent advances in large language models have enabled automated dataset creation at scale. In our VAULT pipeline, we synthesize adversarial hypotheses with Llama-4-Scout-17B-16E-Instruct and then employ an ensemble of Gemma-3-27B-IT, Phi-4, and Qwen3-32B to unanimously validate each candidate before fine-tuning RoBERTa-base (Liu et al., 2019). This approach builds on synthetic data methods such as GNLI (Hosseini et al., 2024), which demonstrated that purely LLM-generated corpora can yield strong zero-shot transfer on ANLI and MNLI, and on counterfactual and paraphrase generation techniques that enrich training distributions (Li et al., 2023; Klemen & Robnik-Šikonja, 2021).

**Retrieval for Few-Shot Prompting.** Quality context examples are critical for reliable generation. Hybrid retrieval-combining semantically rich embeddings (BGE) with robust lexical scoring (BM25)-has been shown to select more diverse and relevant few-shots, leading to higher-fidelity outputs and fewer label errors downstream (Chen et al., 2024; Robertson & Zaragoza, 2009).

**Automated Adversarial Example Generation.** Automated adversarial pipelines seek to stress-test and fortify NLI models without manual curation. (Minervini & Riedel, 2018) generate logical-constraint-violating instances via LLM prompting, improving SNLI and MultiNLI robustness by 2-3% (Minervini & Riedel, 2018). Nie et al.'s ANLI leverages a model-in-the-loop to surface challenging examples, boosting out-of-domain transfer by roughly 5% (Nie et al., 2020). Iyyer et al.'s SCPNs apply controlled syntactic transformations to create paraphrase-based attacks, yielding a 4% robustness gain (Iyyer et al., 2018). More recent work on large-scale synthetic NLI data (e.g. GNLI) has shown that such corpora can rival or surpass real training sets on zero-shot benchmarks (Hosseini et al., 2024). Unlike these prior methods, VAULT fully automates retrieval, adversarial generation, multi-LLM validation, and iterative retraining, providing a scalable, end-to-end solution for enhancing NLI models' resilience.

## 3 METHODOLOGY

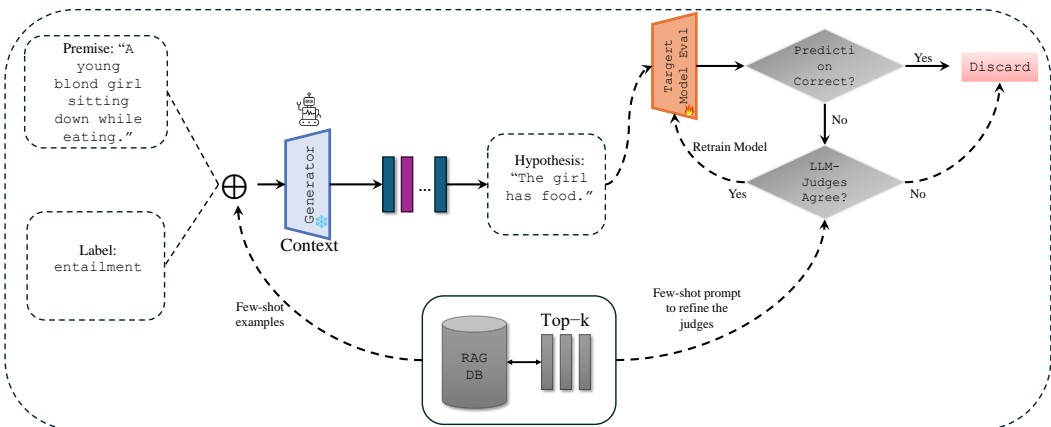

Figure 2: Overview of the VAULT pipeline: combined semantic and lexical retrieval of balanced few-shot contexts, adversarial hypothesis generation via LLM, ensemble validation for label fidelity, and iterative retraining of the NLI model.

Let $\mathcal{D} = \{(p_i, y_i)\}_{i=1}^N$ be the NLI training set on which the target model was trained, where each premise $p_i$ is paired with a label $y_i \in \{\text{entail, neutral, contradict}\}$. We denote by $M^{(t)}$ the target NLI model after $t$ rounds of adversarial retraining, with $M^{(0)}$ pre-trained on $\mathcal{D}$. The VAULT pipeline (see Fig. 1) enhances $M$ through five stages-Retrieval, Hypothesis Generation, Adversarial filtering, Automated Validation, and Iterative Retraining-applied to each $(p, y) \in \mathcal{D}$. Figure 2 provides an overview.

**1. Retrieval** For each premise $p$, we assemble a label-balanced few-shot context $\mathcal{C}_p = \bigcup_{y' \in \{\text{entail, neutral, contradict}\}} \mathcal{C}_p^r(y')$, with $|\mathcal{C}_p| = 3k$ by retrieving $k$ examples per label under mode $r \in \{\text{sem, lex}\}$. Let $\mathcal{D}_{y'}$ be all premises with label $y'$.

**Semantic Retrieval.** We denote the text embedder as $emb$. Embed each premise $x$ as $e_x = E_{\text{emb}}(x) \in \mathbb{R}^d$. For query $p$, $e_p = E_{\text{emb}}(p)$, and for each $y'$ we select

$$\mathcal{C}_p^{\text{sem}}(y') = \arg \max_{\substack{S \subseteq \mathcal{D}_{y'} \\ |S|=k}} \sum_{x \in S} \cos(e_p, e_x),$$

i.e. the $k$ nearest neighbors by cosine similarity in embedding space.

**Lexical (BM25) Retrieval.** Index premises with BM25 ($k_1 = 1.5, b = 0.75$) and define

$$s_{\text{BM25}}(p, x) = \sum_{t \in p} \text{IDF}(t) \cdot \frac{\text{tf}(t, x)(k_1 + 1)}{\text{tf}(t, x) + k_1(1 - b + b\frac{|x|}{\text{avgdl}})}.$$

Then for each $y'$, retrieve

$$\mathcal{C}_p^{\text{lex}}(y') = \arg \max_{\substack{S \subseteq \mathcal{D}_{y'} \\ |S|=k}} \sum_{x \in S} s_{\text{BM25}}(p, x).$$

Combining both modes yields $|\mathcal{C}_p| = 3k$ with equal representation of each NLI class, blending semantic depth and lexical relevance.

**Combined (semantic + BM25) Retrieval.** To leverage both semantic and lexical signals, we first compute for each candidate $x$ and query $p$:

$$\tilde{s}_{\text{sem}}(p, x) = \frac{\cos\big(E_{\text{emb}}(p), E_{\text{emb}}(x)\big) - \mu_{\text{sem}}}{\sigma_{\text{sem}}}, \tag{1}$$

$$\tilde{s}_{\text{lex}}(p, x) = \frac{s_{\text{BM25}}(p, x) - \mu_{\text{lex}}}{\sigma_{\text{lex}}}, \tag{2}$$

$$s_{\text{comb}}(p, x) = \alpha \, \tilde{s}_{\text{sem}}(p, x) + (1 - \alpha) \, \tilde{s}_{\text{lex}}(p, x). \tag{3}$$

where $\mu$ and $\sigma$ are the corpus mean and standard deviation of each score. We then form a weighted sum

$$s_{\text{comb}}(p, x) = \alpha \, \tilde{s}_{\text{sem}}(p, x) + (1 - \alpha) \, \tilde{s}_{\text{lex}}(p, x),$$

with $\alpha \in [0, 1]$ controlling the interpolation between semantic and lexical retrieval. Finally, for each label $y'$, we retrieve

$$\mathcal{C}_p^{\text{comb}}(y') = \arg \max_{\substack{S \subseteq \mathcal{D}_{y'} \\ |S|=k}} \sum_{x \in S} s_{\text{comb}}(p, x),$$

selecting the top-$k$ premises by combined score. This yields $|\mathcal{C}_p| = 3k$ with examples that capture both deep contextual similarity and surface-level overlap.

In each run, we set

$$\mathcal{C}_p = \bigcup_{y' \in \{\text{entail, neutral, contradict}\}} \mathcal{C}_p^r(y'),$$

yielding a balanced prompt context of size $3k$.

As described in Algorithm 1, we retrieve a label-balanced few-shot context for each premise by selecting the top-$k$ examples per NLI class under semantic, lexical, or combined scoring.

---

**Algorithm 1** Balanced Few-Shot Context Retrieval (with combined mode)

---

**Input**: Premise $p$, dataset $\mathcal{D}$ partitioned by label $\{\mathcal{D}_y\}$
**Parameter**: examples per label $k$, mode $r \in \{\mathrm{sem}, \mathrm{lex}, \mathrm{comb}\}$
**Output**: Few-shot context $\mathcal{C}_p$

  1: $\mathcal{C}_p \leftarrow \emptyset$
  2: **if** $r = \mathrm{comb}$ **then**
  3:     compute $\mu_{\mathrm{sem}}, \sigma_{\mathrm{sem}}$ over EMB scores
  4:     compute $\mu_{\mathrm{lex}}, \sigma_{\mathrm{lex}}$ over BM25 scores
  5:     $e_p \leftarrow E_{\mathrm{emb}}(p)$
  6: **end if**
  7: **for** each label $y' \in \{\mathrm{entail}, \mathrm{neutral}, \mathrm{contradict}\}$ **do**
  8:     **for** each $x \in \mathcal{D}_{y'}$ **do**
  9:       **if** $r = \mathrm{sem}$ **then**
10:         $\mathrm{scores}[x] \leftarrow \cos(e_p, E_{\mathrm{emb}}(x))$
11:       **else if** $r = \mathrm{lex}$ **then**
12:         $\mathrm{scores}[x] \leftarrow s_{\mathrm{BM25}}(p, x)$
13:       **else if** $r = \mathrm{comb}$ **then**
14:         $\tilde{s}_{\mathrm{sem}} \leftarrow \frac{\cos(e_p, E_{\mathrm{emb}}(x)) - \mu_{\mathrm{sem}}}{\sigma_{\mathrm{sem}}}$
15:         $\tilde{s}_{\mathrm{lex}} \leftarrow \frac{s_{\mathrm{BM25}}(p, x) - \mu_{\mathrm{lex}}}{\sigma_{\mathrm{lex}}}$
16:         $\mathrm{scores}[x] \leftarrow \alpha\, \tilde{s}_{\mathrm{sem}} + (1 - \alpha)\, \tilde{s}_{\mathrm{lex}}$
17:       **end if**
18:     **end for**
19:     $\mathrm{top\_k} \leftarrow \arg\max^k_{x \in \mathcal{D}_{y'}} \mathrm{scores}[x]$
20:     $\mathcal{C}_p \leftarrow \mathcal{C}_p \cup \mathrm{top\_k}$
21: **end for**
22: **return** $\mathcal{C}_p$

---

**2. Hypothesis Generation**   Given input $(p, \mathcal{C}_p, y)$ from stage 1, we employ a LLM to produce a hypothesis $h$.

**3. Adversarial Filtering**   Once generated, the hypothesis is paired with its premise and label for classification by the target model. Hypotheses correctly classified by the model are discarded, ensuring only the misclassified examples are kept.

**4. Automated Validation**   Let $\mathcal{H}_p = \{h \mid M^{(t)}(p, h) \neq y\}$ be the set of generated candidates. Each $(p, h) \in \mathcal{H}_p$ is validated by three LLM judges - Gemma-3-27B-IT (Google Research, 2025), Phi-4 (Microsoft Research, 2025), and Qwen3-32B (Qwen Team, 2025). Denote each referee's label by $v_j = M_j(p, h)$. We retain $(p, h, y)$ only if all three agree:

$$\sum_{j=1}^{3} \mathbf{1}[v_j = y] = 3.$$

This unanimous-vote check guarantees maximal label fidelity without manual effort.

**5. Iterative Retraining**   At iteration $t$, let $\mathcal{D}_{\mathrm{adv}}^{(t)}$ be the set of validated adversarial triples. We update the training set

$$\mathcal{D}^{(t+1)} = \mathcal{D} \cup \mathcal{D}_{\mathrm{adv}}^{(t)}$$

and fine-tune $M^{(t)}$ (e.g. for $E = 3$ epochs, learning rate $\eta = 2\times10^{-5}$, batch size $B = 32$) to obtain $M^{(t+1)}$. Repeat for $t = 0, \ldots, T - 1$, progressively hardening the model.

This closed-loop process-retrieve, generate, validate (unanimously), retrain-enables VAULT to iteratively strengthen NLI models by exposing them to increasingly challenging, automatically mined adversarial examples.

The overall VAULT pipeline is summarized in Algorithm 2.

---

**Algorithm 2** VAULT Pipeline: Automated Adversarial RAG

---

**Input**: Training set $\mathcal{D} = \{(p_i, y_i)\}_{i=1}^N$, examples per label $k$, iterations $T$, retrieval mode $r$
**Output**: Enhanced model $M^{(T)}$

1: Train initial model $M^{(0)}$ on $\mathcal{D}$
2: **for** $t = 0$ to $T - 1$ **do**
3:    $\mathcal{D}_{\mathrm{adv}} \leftarrow \emptyset$
4:    **for** each $(p, y) \in \mathcal{D}$ **do**
5:       $\mathcal{C}_p \leftarrow \mathrm{RETRIEVE\_CONTEXT}(p, \mathcal{D}, k, r)$
6:       $h \leftarrow \mathrm{GENERATE\_HYPOTHESIS}(p, \mathcal{C}_p, y)$
7:       **if** $M^{(t)}(p, h) \neq y$ **and** $\mathrm{UNANIMOUS\_VALIDATE}(p, h, y)$ **then**
8:          $\mathcal{D}_{\mathrm{adv}} \leftarrow \mathcal{D}_{\mathrm{adv}} \cup \{(p, h, y)\}$
9:       **end if**
10:   **end for**
11:   Fine-tune $M^{(t+1)}$ on $\mathcal{D} \cup \mathcal{D}_{\mathrm{adv}}$
12: **end for**
13: **return** $M^{(T)}$

---

## 3.1 Hyperparameter Tuning for Retrieval

Retrieval quality depends critically on the interpolation between our semantic and lexical similarity scores. To find the optimal combination weight $\alpha$, we perform a grid search on the SNLI training partition (1,000 examples), using BGE M3 as the embedder, a 9-shot prompt context per label, and aggregating scores across three independent judges. We cast each premise-candidate pair $(p, x)$ as a binary relevance decision-positive if $\mathrm{label}(x) = \mathrm{label}(p)$, negative otherwise-and compute the combined score

$$s_{\mathrm{comb}}(p, x) = \alpha\, \tilde{s}_{\mathrm{sem}}(p, x) + (1 - \alpha)\, \tilde{s}_{\mathrm{lex}}(p, x).$$

For each $\alpha \in \{0, 0.01, 0.02, \ldots, 1.0\}$, we evaluate the area under the ROC curve (ROC AUC) across all positive/negative pairs. ROC AUC is a threshold-agnostic ranking metric that quantifies how well $s_{\mathrm{comb}}$ separates relevant from irrelevant examples; we identify $\alpha^* = 0.83$ as the maximizer. At this setting, the ROC curve (Figure 4) achieves an AUC of 0.93. For all downstream experiments, we then fix $\alpha = 0.83$ and set the few-shot size $k = 1$ per label-yielding a prompt context of three examples-to balance context richness with computational efficiency.

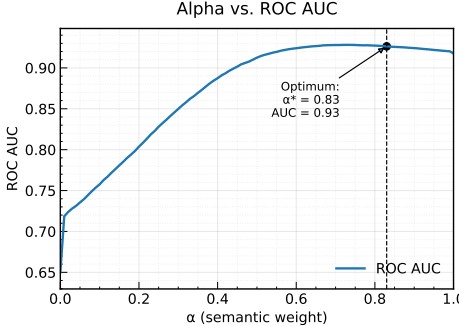 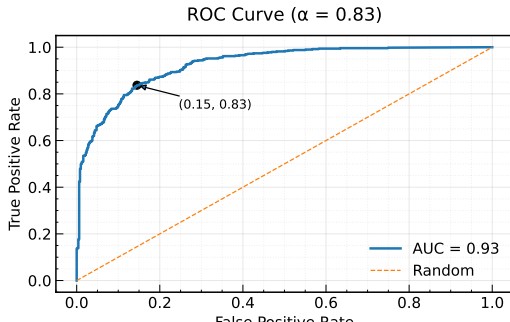

Figure 3: Variation of ROC AUC as a function of the semantic-lexical weighting parameter $\alpha$.

Figure 4: ROC curve at optimal $\alpha = 0.83$ (AUC = 0.93).

## 3.2 Avoiding Forgetness

Fine-tuning a pretrained NLI model solely on adversarial examples $\mathcal{D}_{\mathrm{adv}}$ can induce *catastrophic forgetting*: the model overfits the new distribution and its performance on the original SNLI data $\mathcal{D}_{\mathrm{orig}}$ degrades. To prevent this, we blend original and adversarial data according to a mixing ratio

$$r = \frac{|\mathcal{D}_{\mathrm{orig}}|}{|\mathcal{D}_{\mathrm{adv}}|} \in \left\{ 0,\, 1,\, \frac{1}{2},\, \frac{1}{3},\, \frac{1}{4} \right\},$$

where $r = 0$ indicates training exclusively on $\mathcal{D}_{\mathrm{adv}}$ (i.e., no original data), and $r = \frac{1}{4}$ denotes one original SNLI example for every four adversarial examples.

For each retrieval mode $m \in \{\mathrm{sem}, \mathrm{lex}\}$, we form the augmented training set

$$\mathcal{D}^{(m)}(r) = \mathcal{D}_{\mathrm{orig}} \cup \mathrm{Sample}\!\left( \mathcal{D}_{\mathrm{adv}}^{(m)},\, |\mathcal{D}_{\mathrm{orig}}|/r \right)$$

and fine-tune the model for $T$ iterations to obtain $M_m^{(T)}$. We then evaluate its overall accuracy on the combined SNLI, ANLI, and Multi-NLI benchmarks:

$$A_m(r) = \mathrm{Accuracy}\!\left( M_m^{(T)} \mid \mathcal{D}^{(m)}(r) \right).$$

Figure 5 plots accuracies (after filtering with LLM-judges) of $A_{\mathrm{BGE}}(r)$, $A_{\mathrm{BM25}}(r)$ and $A_{\mathrm{BGE+BM25}}(r)$ as functions of the adversarial-to-original ratio $r$. All three curves climb steeply from $r = 0$ to $r = \frac{1}{2}$, with BGE rising from 90.10% to 92.13% and BM25 from 90.09% to 92.00%. The combined BGE+BM25 strategy consistently outperforms either alone, increasing from 90.54% to 92.33% over the same range. Each curve reaches its maximum at $r = \frac{1}{4}$, where the combined method peaks at 92.60%, indicating that one validated adversarial example per four originals strikes the best balance between robustness and retention. Beyond $r = \frac{1}{4}$, further mixing yields only marginal gains.

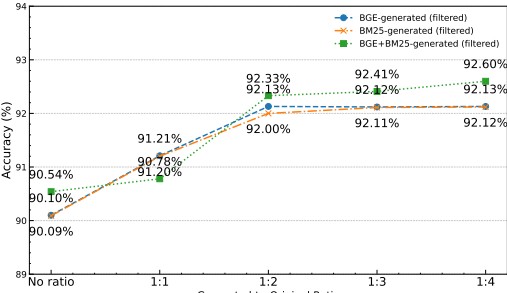

Figure 5: Overall accuracy $A_m(r)$ on SNLI, ANLI, and Multi-NLI versus mixing ratio $r$ of generated adversarial examples to original SNLI, for BGE (blue, dashed) and BM25 (orange, dash-dot) pipelines.

By selecting $r^* = \frac{1}{4}$, we effectively mitigate catastrophic forgetting-preserving SNLI performance-while still reaping substantial adversarial robustness gains. This controlled mixing injects diversity into the training distribution and produces models that generalize reliably across both original and adversarial scenarios.

## 4 EVALUATION SETUP: MODELS AND DATASETS

To evaluate the effectiveness of our adversarial RAG pipeline, we fine-tune and test a suite of models on three standard NLI benchmarks:

- **Target NLI Model:** `RoBERTa-base-SNLI` (125M parameters) (HuggingFace, 2022), a RoBERTa variant pretrained on SNLI.
- **Generation LLM:** Adversarial hypotheses are generated with `Llama-4-Scout-17B-16E-Instruct` (Meta AI, 2025).
- **Validation LLMs:** Each candidate pair is vetted by an ensemble of three models:

- Gemma-3-27B-IT (Google Research, 2025),
- Phi-4 (Microsoft Research, 2025), and
- Qwen3-32B (Qwen Team, 2025).

We report results on the following **NLI** test sets: **SNLI test** (Bowman et al., 2015), the original human-annotated NLI benchmark; **ANLI** (Nie et al., 2019), featuring adversarial examples collected via human-in-the-loop attacks; and **Multi-NLI** (Williams et al., 2018), a multi-genre corpus for evaluating cross-domain generalization.

## 5 RESULTS

Table 1: Accuracy (%) of RoBERTa-base on each test set in the zero-shot setup under adversarial mixing. Method names list only *BGE*, *BM25*, and *BGE+BM25*, denoting their respective generation methods. "Filtered" indicates unanimous LLM validation.

| Dataset | RoBERTa-Base | Additional Data | Para-phrasing | GNLI | Method | Filtered | $r=0$ | $r=$ 1:1 | $r=$ 1:2 | $r=$ 1:3 | $r=$ 1:4 |
|---|---|---|---|---|---|---|---|---|---|---|---|
| | | | | – | BGE | No | 90.98% | 91.17% | 91.51% | 91.54% | 91.55% |
| | | | | – | BGE | Yes | 90.10% | 91.21% | 92.13% | 92.12% | 92.13% |
| | | | | – | BM25 | No | 90.03% | 91.02% | 91.14% | 91.18% | 91.19% |
| | | | | – | BM25 | Yes | 90.09% | 91.20% | 92.00% | 92.11% | 92.12% |
| SNLI | 88.48% | 89.42% | 84.73% | – | BGE+BM25 | No | 90.11% | 91.19% | 91.35% | 91.61% | 91.68% |
| | | | | – | BGE+BM25 | Yes | 90.54% | 90.78% | 92.33% | 92.41% | **92.60%** |
| | | | | – | T5-Small | - | - | - | - | - | - |
| | | | | – | T5-Large | - | - | - | - | - | - |
| | | | | – | T5-XXL | - | - | - | - | - | - |
| | | | | – | BGE | No | 79.07% | 79.72% | 79.52% | 79.92% | 79.47% |
| | | | | – | BGE | Yes | 78.72% | 79.12% | 80.02% | 79.72% | 80.27% |
| | | | | – | BM25 | No | 78.07% | 78.52% | 78.72% | 78.82% | 78.88% |
| Adversarial NLI | 75.04% | 77.07% | 72.39% | – | BM25 | Yes | 77.97% | 78.62% | 78.92% | 79.07% | 79.12% |
| | | | | – | BGE+BM25 | No | 78.11% | 79.18% | 78.51% | 78.99% | 78.91% |
| | | | | – | BGE+BM25 | Yes | 79.12% | 80.43% | 80.67% | 80.89% | **80.95%** |
| | | | | 33.00% | T5-Small | - | - | - | - | - | - |
| | | | | 45.72% | T5-Large | - | - | - | - | - | - |
| | | | | 57.87% | T5-XXL | - | - | - | - | - | - |
| | | | | – | BGE | No | 69.54% | 69.32% | 70.22% | 69.72% | 71.08% |
| | | | | – | BGE | Yes | 69.15% | 69.62% | 70.22% | 69.97% | 71.15% |
| | | | | – | BM25 | No | 68.34% | 68.72% | 68.92% | 69.22% | 69.54% |
| MultiNLI | 54.67% | 57.61% | 50.01% | – | BM25 | Yes | 68.57% | 68.82% | 69.12% | 69.47% | 69.74% |
| | | | | – | BGE+BM25 | No | 68.05% | 68.15% | 68.81% | 69.11% | 70.02% |
| | | | | – | BGE+BM25 | Yes | 69.21% | 69.37% | 70.59% | 69.81% | **71.99%** |
| | | | | 82.18% | T5-Small | - | - | - | - | - | - |
| | | | | 90.61% | T5-Large | - | - | - | - | - | - |
| | | | | **91.77%** | T5-XXL | - | - | - | - | - | - |

Table 2: Few-shot accuracy (%) of our generation methods on each test set. Columns indicate the number of few-shot examples; the 6-shot column reproduces the $r=1:4$ results from Table 1. Bold indicates the best performance per row.

| Dataset | Method | Filtered? | 0-shot | 3-shot | 6-shot | 9-shot |
|---|---|---|---|---|---|---|
| | BGE | No | 87.51% | 90.05% | 91.55% | **91.56%** |
| | BGE | Yes | 88.18% | 90.69% | 92.13% | **92.15%** |
| SNLI | BM25 | No | 87.51% | 89.69% | 91.19% | 91.22% |
| | BM25 | Yes | 88.18% | 90.67% | **92.12%** | 92.11% |
| | BGE+BM25 | No | 87.51% | 89.98% | **91.68%** | 91.51% |
| | BGE+BM25 | Yes | 88.18% | 90.71% | **92.60%** | 92.51% |
| | BGE | No | 75.81% | 77.72% | 79.47% | **79.47%** |
| | BGE | Yes | 76.27% | 78.76% | **80.27%** | 80.26% |
| Adversarial NLI | BM25 | No | 75.81% | 77.37% | 78.87% | **78.88%** |
| | BM25 | Yes | 76.27% | 77.60% | **79.12%** | 79.10% |
| | BGE+BM25 | No | 75.81% | 77.71% | 78.81% | **78.91%** |
| | BGE+BM25 | Yes | 76.27% | 77.81% | 78.95% | **80.95%** |
| | BGE | No | 67.18% | 69.25% | 71.07% | **71.08%** |
| | BGE | Yes | 67.87% | 69.02% | **71.12%** | 71.15% |
| MultiNLI | BM25 | No | 67.18% | 68.07% | **69.57%** | 69.54% |
| | BM25 | Yes | 67.87% | 68.22% | 69.72% | **69.74%** |
| | BGE+BM25 | No | 67.18% | 69.42% | 69.99% | **70.02%** |
| | BGE+BM25 | Yes | 67.87% | 71.00% | 71.12% | **71.99%** |

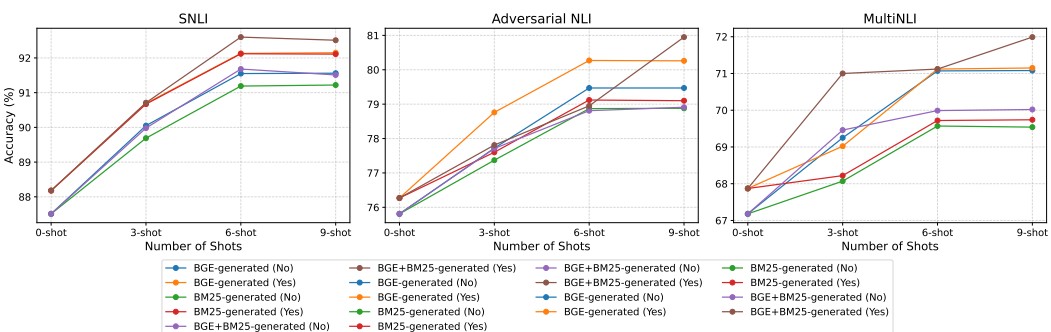

Figure 6: Few-shot accuracy of generation methods by dataset.

To contextualize these gains, we compare against models fine-tuned solely on GNLI (Hosseini et al., 2024), a synthetic NLI corpus of ~685K LLM-generated examples. With only GNLI, RoBERTa-Base reaches 89.42% on SNLI, 77.07% on ANLI, and 57.61% on MultiNLI. Our VAULT pipeline generates ~30K adversarial candidates per retrieval strategy, retaining 6637 BGE and 5991 BM25 after human validation. Injecting these at a 1:4 ratio boosts RoBERTa-Base from 88.48% to 92.60% on SNLI, from 75.04% to 80.95% on ANLI, and from 54.67% to 71.99% on MultiNLI (Table 1). Table 1 also shows unfiltered data reaching 91.55% on SNLI at ($r = \frac{1}{4}$), with filtering improving further. Overall, a small, validated adversarial set, particularly from BGE, outperforms massive, untargeted corpora across benchmarks. Table 2 and Figure 6 show few-shot accuracy of BGE and BM25 (with/without filtering) on SNLI, ANLI, and MultiNLI. On SNLI, unfiltered BGE rises from 87.51% (0-shot) to 91.56% (9-shot), filtered BGE from 88.18% to 92.15%; unfiltered BM25 from 87.51% to 91.22%, filtered BM25 from 88.18% to 92.11%. On ANLI, filtered BGE improves from 76.27% to 80.26%, unfiltered BM25 peaks at 78.88%. On MultiNLI, filtered BGE grows from 67.87% to 71.15%, unfiltered BM25 tops at 69.54% (filtered 67.87% to 69.74%). Notably, 6-shot matches or exceeds the 1:4 mixing results in Table 1.

## 6 CONCLUSION

In this work, we introduce **VAULT**, an adversarially-driven data augmentation framework that cuts reliance on massive synthetic corpora while matching-or often surpassing-state-of-the-art performance. Rather than fine-tuning on hundreds of thousands of examples (e.g. 685 K in GNLI), VAULT generates ~30 K adversarial candidates per retrieval strategy, validates them, and retains just 6-6.6 K samples. This lean approach yields a 4-7 point gain over GNLI-only baselines in zero- and few-shot settings on SNLI, ANLI, and MultiNLI despite using an order of magnitude less data. TF-IDF and BERTScore analyses confirm these focused sets preserve both lexical overlap and semantic fidelity. In future work, we'll explore automated validation heuristics, extensions to other NLI domains, and combinations of adversarial augmentation with model-centric techniques for even greater efficiency.

## 7 LIMITATIONS AND FUTURE WORK

While VAULT shows strong gains with minimal synthetic data, it relies on large-scale LLMs for generation (Llama-4-Scout-17B-16E-Instruct) and validation (Gemma-3-27B-IT, Phi-4, Qwen3-32B), incurring notable compute costs that may limit use in resource-constrained settings. The unanimous-agreement filter, while ensuring high-quality examples, may discard borderline cases that could diversify training. Our experiments focus on English NLI; extending to multilingual or specialized domains may require adapting retrieval strategies and validation ensembles. Future work includes exploring lighter validation (e.g., smaller ensembles or learned filters), adaptive retrieval budgets, automated threshold calibration, continual learning where candidates are generated on the fly, and integration with model-centric robustness methods such as contrastive fine-tuning and adversarial regularization.

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

## A   APPENDIX

### A.1   JUDGE ENSEMBLE CONFIGURATION

With the retrieval weight fixed at $\alpha = 0.83$ and the generated-to-original example ratio set to 1:4, we evaluated the impact of varying the number of "judges" (independent LLM validators) on downstream accuracy. All experiments were run on the SNLI test set. We filtered examples by requiring unanimous agreement among the selected judges and then measured classification accuracy on the remaining items.

Table 3:   Filtering and accuracy under different judge ensemble sizes (SNLI test, 1:4 gen:orig, $\alpha = 0.83$). Judges: G = Gemma-3-27B-IT (Google Research, 2025), Q = Qwen3-32B (Qwen Team, 2025), P = Phi-4 (Microsoft Research, 2025).

| # Judges | # Examples | Accuracy (%) | Judges |
|---|---|---|---|
| 1 | 16,147 | 91.02 | G |
| 2 | 9,312 | 91.49 | G + Q |
| 3 | 6,438 | **92.13** | G + Q + P |

As shown in Table 3 and Figure 7, the three-judge ensemble yields the highest accuracy (92.13%) on 6,438 filtered observations. Both the two-judge and single-judge configurations retain more examples but achieve lower accuracies of 91.49% (9,312 examples) and 91.02% (16,147 examples), respectively. Gemma-3-27B-IT consistently remains in all configurations, with Qwen3-32B joining for the two-judge setup and Phi-4 for the three-judge ensemble. We adopt the three-judge configuration for all subsequent evaluations.

### A.2   DATASET COMPARISON

To gain insights into the relationship between the data generated in our experiment and existing benchmarks, we first extracted the 10 most frequent non-stopwords from each dataset. This qualitative analysis highlights topical overlap and domain shifts. To quantify similarity more rigorously, we computed two complementary metrics across seven collections-SNLI Train, BGE-generated, BM25-generated, SNLI Test, Adversarial NLI, Multi-NLI, and our combined BGE+BM25-generated set: TF-IDF cosine similarity and BERTScore F1 (Zhang et al., 2019).

**TF-IDF Cosine Similarity.** Let each dataset $D$ be represented by a TF-IDF vector $\mathbf{v}_D \in \mathbb{R}^n$, where $n$ is the vocabulary size and the $i$th component is

$$v_{D,i} = \mathrm{TF}_{D,i} \cdot \log\left(\frac{N}{\mathrm{DF}_i}\right),$$

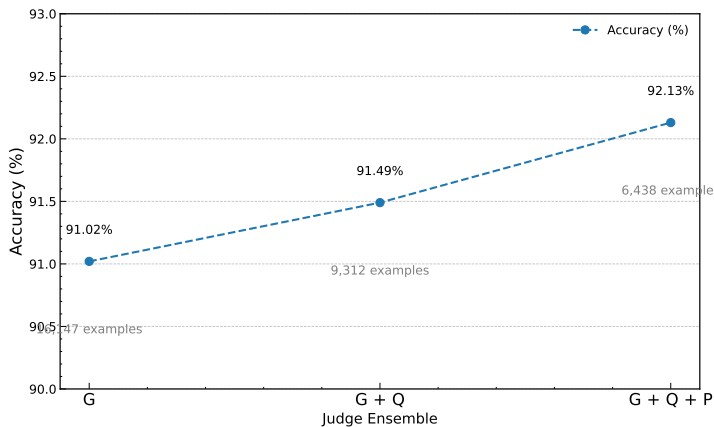

Figure 7: Accuracy vs. number of judges (SNLI test, $\alpha = 0.83$, 1:4 generated:original). Points are annotated with the number of filtered examples.

with $\text{TF}_{D,i}$ the term frequency in $D$, $N$ the total number of datasets, and $\text{DF}_i$ the number of datasets containing term $i$. We then define

$$\text{sim}_{\text{TFIDF}}(D, D') = \frac{\mathbf{v}_D \cdot \mathbf{v}_{D'}}{\|\mathbf{v}_D\| \, \|\mathbf{v}_{D'}\|}.$$

Figure 8 shows the resulting $7 \times 7$ matrix. Notably, the combined BGE+BM25 set has a TF-IDF similarity of approximately 0.0251 with SNLI Train, 0.0188 with SNLI Test, and 0.0150 with Multi-NLI-intermediate between its BGE-only and BM25-only counterparts.

**BERTScore F1.** We next measure semantic overlap by applying BERTScore F1, which aligns token embeddings from a pre-trained transformer and computes an $F_1$ score:

$$\text{P} = \frac{1}{|x|} \sum_{t \in x} \max_{s \in y} \cos(\mathbf{e}_t, \mathbf{e}_s), \text{R} = \frac{1}{|y|} \sum_{s \in y} \max_{t \in x} \cos(\mathbf{e}_s, \mathbf{e}_t),$$

$$\text{F1} = 2 \cdot \frac{\text{P} \, \text{R}}{\text{P} + \text{R}},$$

where $x, y$ are token sequences from two datasets and $\mathbf{e}$ are contextual embeddings. Figure 9 displays the $7 \times 7$ BERTScore F1 matrix. The combined set scores about 0.8658 with SNLI Train, 0.8534 with SNLI Test, 0.8458 with Adversarial NLI, and 0.8554 with Multi-NLI, again falling between its BGE-only and BM25-only pairs. These results confirm that our validated adversarial examples share both lexical and semantic patterns with standard NLI benchmarks, while still introducing novel, challenging variations.

From Figure 8, we see that both BGE- and BM25-generated data share moderate lexical overlap with the original SNLI Train set (cosine similarities around 0.02-0.03), but diverge more substantially from the Adversarial NLI and Multi-NLI benchmarks. In contrast, Figure 9 shows that semantically these generated datasets align much more closely with SNLI Train and SNLI Test (BERTScore F1 values above 0.85), indicating that although the surface vocabulary varies, the core contextual meaning is well preserved.

### A.3 GENERATED DATASET CHARACTERISTICS AND HYPOTHESIS LENGTHS

We first examined the most frequent tokens in each corpus to identify thematic patterns. In the `SNLI train` (Bowman et al., 2015) and `SNLI test` (Bowman et al., 2015) sets, words like "man," "woman," and "people" dominate, reflecting descriptions of social interactions. The `Adversarial NLI` dataset (Nie et al., 2019) shifts focus to media and chronology, with top tokens such as "film," "first," and "scene," while the `Multi-NLI test` set (Williams et al., 2018) uses more abstract, domain-diverse language-terms like "author," "context," and "claim" appear frequently.

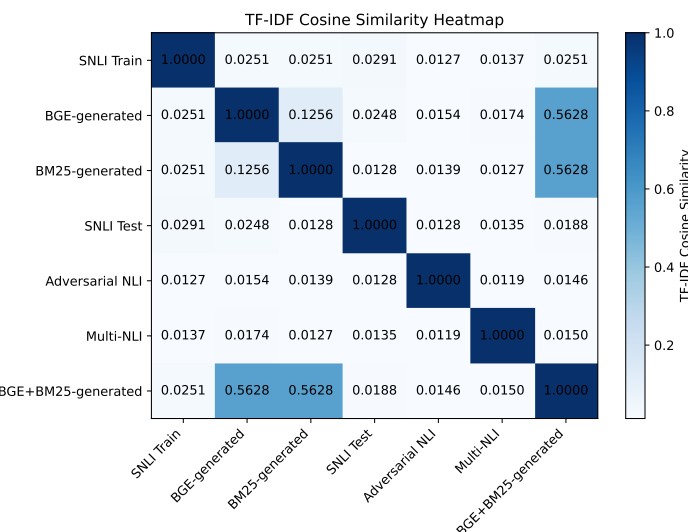

Figure 8: Pairwise TF-IDF cosine similarity between datasets.

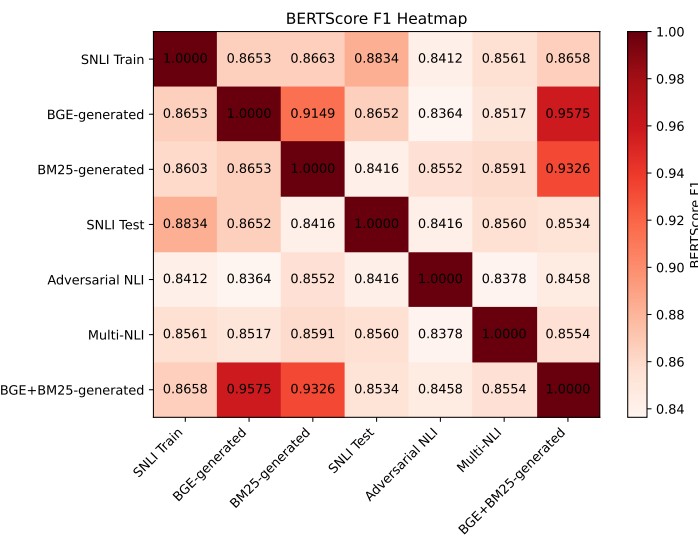

Figure 9: Pairwise BERTScore F1 between datasets.

Turning to our three LLM-generated sets-`Generated-BM25`, `Generated-BGE` and `BGE+BM25`-we again see a high incidence of speculative and gender-related terms ("could," "would," "woman," "he," "she"), confirming that all retrieval strategies surface similar thematic content with only minor stylistic differences.

Figure 10 compares the average hypothesis lengths across all seven datasets. Each of the generated sets produces the longest hypotheses-around 98-100 characters (16-17 words)-demonstrating the LLM's tendency toward more elaborate constructions when given rich few-shot contexts. By contrast, the `SNLI train` and `SNLI test` annotations remain quite concise ($\approx$ 37-38 characters, 7-8 words), reflecting the brevity of human-written examples. The `Adversarial NLI` instances average $\approx$ 64 characters (11 words), and the `Multi-NLI` examples average $\approx$ 56 characters (10 words), underscoring their intermediate complexity. These length patterns highlight how our adversarial RAG pipeline generates richer, more challenging hypotheses while preserving diversity across data sources.

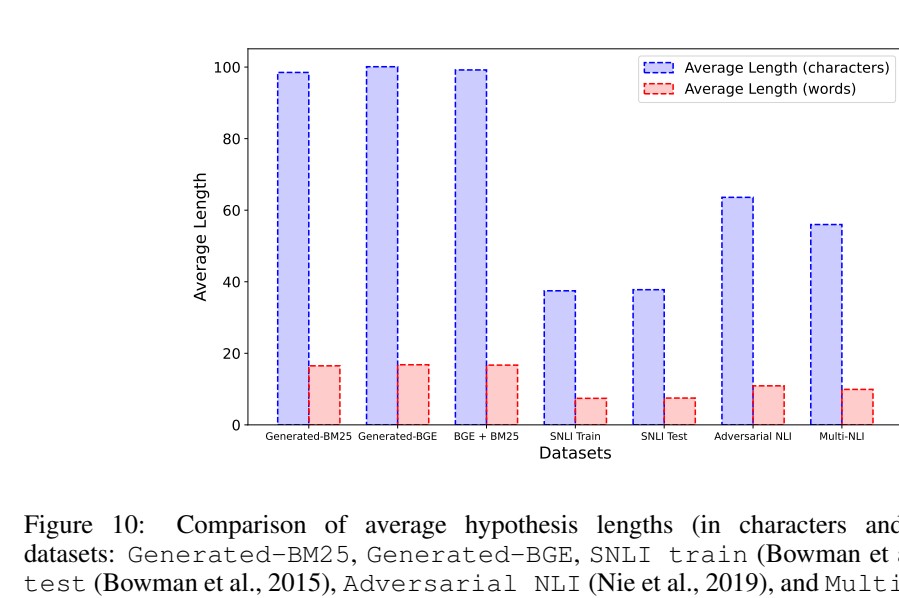

Figure 10: Comparison of average hypothesis lengths (in characters and words) across datasets: `Generated-BM25`, `Generated-BGE`, `SNLI train` (Bowman et al., 2015), `SNLI test` (Bowman et al., 2015), `Adversarial NLI` (Nie et al., 2019), and `Multi-NLI` (Williams et al., 2018).

### A.4 RETRIEVAL ACCURACY ACROSS SIMILARITY METRICS

For purely lexical retrieval we employ BM25 with parameters $k_1 = 1.5$ and $b = 0.75$. The BM25 score for a query $p$ and document $x$ is given by

$$s_{\text{BM25}}(p, x) = \sum_{t \in p} \text{IDF}(t) \frac{\text{tf}(t, x)(k_1 + 1)}{\text{tf}(t, x) + k_1\left(1 - b + b\frac{|x|}{\text{avgdl}}\right)}, \tag{4}$$

and for each label $y'$ we retrieve the top-$k$ documents

$$\mathcal{C}_p^{\text{lex}}(y') = \arg \max_{\substack{S \subseteq \mathcal{D}_{y'} \\ |S| = k}} \sum_{x \in S} s_{\text{BM25}}(p, x). \tag{5}$$

For embedding-based retrieval, we first compute cosine similarity

$$S_{\cos}(E_I, E_\mathcal{D}) = \frac{E_I \cdot E_\mathcal{D}}{\|E_I\|_2 \|E_\mathcal{D}\|_2}, \tag{6}$$

and raw dot product

$$S_{\text{dp}}(E_I, E_\mathcal{D}) = E_I \cdot E_\mathcal{D} = \sum_{i=1}^{d} (E_I)_i (E_\mathcal{D})_i. \tag{7}$$

We additionally assess two norm-based distances: the $L_2$ distance

$$d_2(E_I, E_\mathcal{D}) = \|E_I - E_\mathcal{D}\|_2 = \sqrt{\sum_{i=1}^{d} \left((E_I)_i - (E_\mathcal{D})_i\right)^2}, \tag{8}$$

and the $L_1$ distance

$$d_1(E_I, E_\mathcal{D}) = \|E_I - E_\mathcal{D}\|_1 = \sum_{i=1}^{d} \left|(E_I)_i - (E_\mathcal{D})_i\right|. \tag{9}$$

Finally, to capture distributional discrepancies we examine the Bray-Curtis distance

$$d_{\text{BC}}(E_I, E_\mathcal{D}) = \frac{\sum_{i=1}^{d} \left|(E_I)_i - (E_\mathcal{D})_i\right|}{\sum_{i=1}^{d} \left|(E_I)_i + (E_\mathcal{D})_i\right|}, \tag{10}$$

and the Canberra distance

$$d_{\mathrm{Can}}(E_I, E_{\mathcal{D}}) = \sum_{i=1}^{d} \frac{\left|(E_I)_i - (E_{\mathcal{D}})_i\right|}{\left|(E_I)_i\right| + \left|(E_{\mathcal{D}})_i\right|}. \tag{11}$$

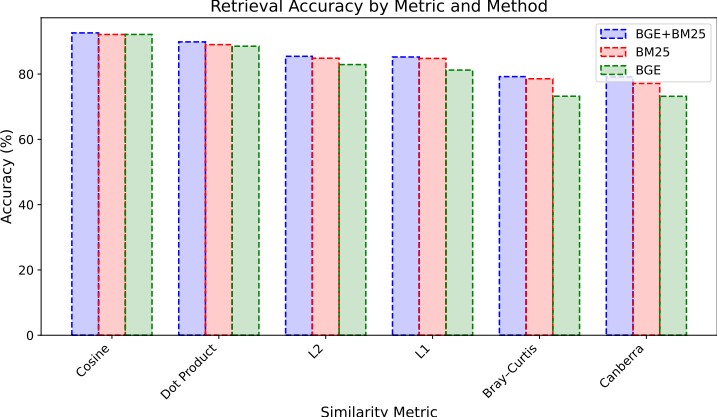

Figure 11: Retrieval accuracy (%) by similarity metric for BGE+BM25, BM25, and BGE.

Figure 11 demonstrates that BGE+BM25 outperforms both BM25 alone and BGE alone across all six metrics, achieving $92.60\%$ (cosine), $89.85\%$ (dot product), $85.43\%$ ($L_2$), $85.22\%$ ($L_1$), $79.21\%$ (Bray-Curtis) and $79.12\%$ (Canberra). Pure BM25 and pure BGE match closely on cosine but degrade more sharply on norm- and distribution-based distances, confirming the robustness of the hybrid lexical-semantic approach.

## A.5 EXAMPLE - FEW-SHOT CHAT SEQUENCE

**BGE based retrieval**    The chat sequences below present a clear few-shot retrieval sequence for a natural language inference task. They illustrate six premise-hypothesis pairs-two each for entailment, neutral, and contradiction-and conclude with a concise model prompt. This format makes the example selection process transparent and highlights the model's reasoning in a single, easily readable block. These examples are based solely on BGE retrieval.

**Example 1: Few-Shot Retrieval & Model Return**

**Shot 1**
**Premise**: A blond little girl enjoying a burrito.
**Label**: entailment.
**Hypothesis:** The girl ate a burrito.

**Shot 2**
**Premise**: A young blond girl sitting down while eating.
**Label**: entailment.
**Hypothesis:** The girl has food.

**Shot 3**
**Premise**: A blond little girl enjoying a burrito.
**Label**: neutral.
**Hypothesis:** The hungry girl ate a burrito at the restaurant.

**Shot 4**
**Premise**: A young blond girl sitting down while eating.
**Label**: neutral.
**Hypothesis:** The girl is eating at a picnic.

**Shot 5**
**Premise**: A blond little girl enjoying a burrito.
**Label**: contradiction.
**Hypothesis:** The brunette girl didn't like the burrito.

**Shot 6**
**Premise**: A young blond girl sitting down while eating.
**Label**: contradiction.
**Hypothesis:** The girl runs all over her house while eating because she can never sit down.

**Llama Generation**
**User:** Now generate a one-sentence hypothesis that contradicts the premise above. Return only the hypothesis without narration.
**Assistant (Llama):** The girl threw the sandwich away.

**Example 2: Few-Shot Retrieval & Model Return**

**Shot 1**
**Premise**: An older man is drinking orange juice at a restaurant.
**Label**: entailment.
**Hypothesis:** A man is drinking juice.

**Shot 2**
**Premise**: An older man is drinking orange juice at a restaurant.
**Label**: entailment.
**Hypothesis:** A man is drinking juice.

**Shot 3**
**Premise**: An older man sits with his orange juice at a small table in a coffee shop while employees in bright colored shirts smile in the background.
**Label**: neutral.
**Hypothesis:** An elderly man sits in a small shop.

**Shot 4**
**Premise**: An older man sits with his orange juice at a small table in a coffee shop while employees in bright colored shirts smile in the background.
**Label**: neutral.
**Hypothesis:** An older man drinks his juice as he waits for his daughter to get off work.

**Shot 5**
**Premise**: An older man sits with his orange juice at a small table in a coffee shop while employees in bright colored shirts smile in the background.
**Label**: contradiction.
**Hypothesis:** A boy flips a burger.

**Shot 6**
**Premise**: An older man is drinking orange juice at a restaurant.
**Label**: contradiction.
**Hypothesis:** Two women are at a restaurant drinking wine.

**Llama Generation**
**User:** Now generate a one-sentence hypothesis that contradicts the premise above. Return only the hypothesis without narration.
**Assistant (Llama):** The coffee shop is empty.

**Example 3: Few-Shot Retrieval & Model Return**

**Shot 1**
**Premise**: A few people in a restaurant setting, one of them is drinking orange juice.
**Label**: entailment.
**Hypothesis:** The diners are at a restaurant.

**Shot 2**
**Premise**: A few people in a restaurant setting, one of them is drinking orange juice.
**Label**: entailment.
**Hypothesis:** The diners are at a restaurant.

**Shot 3**
**Premise**: A few people in a restaurant setting, one of them is drinking orange juice.
**Label**: neutral.
**Hypothesis:** The people are eating omelettes.

**Shot 4**
**Premise**: A few people in a restaurant setting, one of them is drinking orange juice.
**Label**: neutral.
**Hypothesis:** The people are eating omelettes.

**Shot 5**
**Premise**: A few people in a restaurant setting, one of them is drinking orange juice.
**Label**: contradiction.
**Hypothesis:** The people are sitting at desks in school.

**Shot 6**
**Premise**: A few people are observing the orange juice section at the grocery store.
**Label**: contradiction.
**Hypothesis:** The people are at a baseball stadium.

**Llama Generation**
**User:** Now generate a one-sentence hypothesis that contradicts the premise above. Return only the hypothesis without narration.
**Assistant (Llama):** The people are at a concert.

**Optimized BGE + BM25 Retrieval with Tuned Alpha** The paragraph below describes an optimized retrieval approach that combines semantic search using BGE embeddings with lexical scoring via BM25. By introducing a tunable alpha parameter, the method linearly interpolates between embedding-based similarity and keyword matching-enabling a dynamic balance that maximizes relevance across diverse query types. Through cross-validation, the optimal alpha value was determined to achieve superior retrieval performance compared to either technique alone.

---

**Example 1: Few-Shot Retrieval & Model Return**

**Shot 1**
**Premise**: A person on a horse jumps over a broken down airplane.
**Label**: entailment.
**Hypothesis:** A person is outdoors, on a horse.

**Shot 2**
**Premise**: A person on a horse jumps over a broken down airplane.
**Label**: entailment.
**Hypothesis:** A person is outdoors, on a horse.

**Shot 3**
**Premise**: A person on a horse jumps over a broken down airplane.
**Label**: neutral.
**Hypothesis:** A person is training his horse for a competition.

**Shot 4**
**Premise**: A person is riding a horse, which is jumping over a grounded airplane.
**Label**: neutral.
**Hypothesis:** The horse jumps over a toy airplane.

**Shot 5**
**Premise**: A person on a horse jumps over a broken down airplane.
**Label**: contradiction.
**Hypothesis:** A person is at a diner, ordering an omelette.

**Shot 6**
**Premise**: A person on a horse jumps over a broken down airplane.
**Label**: contradiction.
**Hypothesis:** A person is at a diner, ordering an omelette.

**Llama Generation**
**User:** Now generate a one-sentence hypothesis that is neutral with the premise above. Return only the hypothesis without narration.
**Assistant (Llama):** The horse is very well trained.

**Example 2: Few-Shot Retrieval & Model Return**

**Shot 1**
**Premise**: Children smiling and waving at camera.
**Label**: entailment.
**Hypothesis:** There are children present.

**Shot 2**
**Premise**: Children smiling and waving at camera.
**Label**: entailment.
**Hypothesis:** There are children present.

**Shot 3**
**Premise**: Children smiling and waving at camera.
**Label**: neutral.
**Hypothesis:** They are smiling at their parents.

**Shot 4**
**Premise**: Children smiling and waving at camera.
**Label**: neutral.
**Hypothesis:** They are smiling at their parents.

**Shot 5**
**Premise**: Children smiling and waving at camera.
**Label**: contradiction.
**Hypothesis:** The kids are frowning.

**Shot 6**
**Premise**: Children smiling and making faces to the camera.
**Label**: contradiction.
**Hypothesis:** The children are deceased.

**Llama Generation**
**User:** Now generate a one-sentence hypothesis that contradicts the premise above. Return only the hypothesis without narration.
**Assistant (Llama):** The children are asleep.

**Example 3: Few-Shot Retrieval & Model Return**

**Shot 1**
**Premise**: An older man is drinking orange juice at a restaurant.
**Label**: entailment.
**Hypothesis:** A man is drinking juice.

**Shot 2**
**Premise**: An older man is drinking orange juice at a restaurant.
**Label**: entailment.
**Hypothesis:** A man is drinking juice.

**Shot 3**
**Premise**: An older man is drinking orange juice at a restaurant.
**Label**: neutral.
**Hypothesis:** A man in a restaurant is waiting for his meal to arrive.

**Shot 4**
**Premise**: An elderly man is drinking orange juice at a cafe.
**Label**: neutral.
**Hypothesis:** An older gentleman is enjoying his orange juice at a new cafe.

**Shot 5**
**Premise**: An older man is drinking orange juice at a restaurant.
**Label**: contradiction.
**Hypothesis:** Two women are at a restaurant drinking wine.

**Shot 6**
**Premise**: An older man is drinking orange juice at a restaurant.
**Label**: contradiction.
**Hypothesis:** Two women are at a restaurant drinking wine.

**Llama Generation**
**User:** Now generate a one-sentence hypothesis that is neutral with the premise above. Return only the hypothesis without narration.
**Assistant (Llama):** The restaurant has a lot of tables.

**BM25 based retrieval**   The retrieval process relies exclusively on the BM25 algorithm, which computes scores based on term frequency, document frequency, and document length normalization. By focusing on exact keyword matches and weighting rarer terms more heavily, BM25 effectively ranks documents that share the most relevant vocabulary with the query. This approach is well-suited for tasks where precise lexical overlap is paramount and semantic generalization is less critical. However, it may miss contextually related content when synonyms or paraphrases are used.

---

**Example 1: Few-Shot Retrieval & Model Return**

**Shot 1**
**Premise**: An older man is drinking orange juice at a restaurant.
**Label**: entailment.
**Hypothesis:** A man is drinking juice.

**Shot 2**
**Premise**: An elderly man is drinking orange juice at a cafe.
**Label**: entailment.
**Hypothesis:** An old man is enjoying a beverage at a cafe.

**Shot 3**
**Premise**: An older man is drinking orange juice at a restaurant.
**Label**: neutral.
**Hypothesis:** A man in a restaurant is waiting for his meal to arrive.

**Shot 4**
**Premise**: An older man is drinking orange juice at a restaurant.
**Label**: neutral.
**Hypothesis:** A man in a restaurant is waiting for his meal to arrive.

**Shot 5**
**Premise**: An older man is drinking orange juice at a restaurant.
**Label**: contradiction.
**Hypothesis:** Two women are at a restaurant drinking wine.

**Shot 6**
**Premise**: An older man is drinking orange juice at a restaurant.
**Label**: contradiction.
**Hypothesis:** Two women are at a restaurant drinking wine.

**Llama Generation**
**User:** Now generate a one-sentence hypothesis that entails the premise above. Return only the hypothesis without narration.
**Assistant (Llama):** A man is drinking orange juice at a restaurant.

---

**Example 2: Few-Shot Retrieval & Model Return**

**Shot 1**
**Premise**: An older man is drinking orange juice at a restaurant.
**Label**: entailment.
**Hypothesis:** A man is drinking juice.

**Shot 2**
**Premise**: An older man is drinking orange juice at a restaurant.
**Label**: entailment.
**Hypothesis:** A man is drinking juice.

**Shot 3**
**Premise**: An older man is drinking orange juice at a restaurant.
**Label**: neutral.
**Hypothesis:** A man in a restaurant is waiting for his meal to arrive.

**Shot 4**
**Premise**: An older man is drinking orange juice at a restaurant.
**Label**: neutral.
**Hypothesis:** A man in a restaurant is waiting for his meal to arrive.

**Shot 5**
**Premise**: An older man is drinking orange juice at a restaurant.
**Label**: contradiction.
**Hypothesis:** Two women are at a restaurant drinking wine.

**Shot 6**
**Premise**: An elderly man is drinking orange juice at a cafe.
**Label**: contradiction.
**Hypothesis:** An elderly man is drinking apple juice at a bar.

**Llama Generation**
**User:** Now generate a one-sentence hypothesis that contradicts the premise above. Return only the hypothesis without narration.
**Assistant (Llama):** The man is drinking coffee.

---

**Example 3: Few-Shot Retrieval & Model Return**

**Shot 1**
**Premise**: A woman in a green jacket and hood over her head looking towards a valley.
**Label**: entailment.
**Hypothesis:** The woman is wearing green.

**Shot 2**
**Premise**: A woman in a green jacket and hood over her head looking towards a valley.
**Label**: entailment.
**Hypothesis:** The woman is wearing green.

**Shot 3**
**Premise**: A woman in a green jacket and hood over her head looking towards a valley.
**Label**: neutral.
**Hypothesis:** The woman is cold.

**Shot 4**
**Premise**: A woman gazes over the valley below.
**Label**: neutral.
**Hypothesis:** she looks at the valley she owns.

**Shot 5**
**Premise**: A woman in a green jacket and hood over her head looking towards a valley.
**Label**: contradiction.
**Hypothesis:** The woman is nake.

**Shot 6**
**Premise**: A woman in a green jacket and hood over her head looking towards a valley.
**Label**: contradiction.
**Hypothesis:** The woman is nake.

**Llama Generation**
**User:** Now generate a one-sentence hypothesis that is neutral with the premise above. Return only the hypothesis without narration.
**Assistant (Llama):** The woman is looking at something.

---

## LLM USAGE DISCLOSURE

In accordance with the ICLR 2026 policy on LLM usage, we disclose that a large language model was utilized during the preparation of this manuscript. The use of the LLM was strictly limited to correcting grammar and checking code syntax. All research contributions, experimental design, analysis, and the core text were generated by the human authors. The authors have reviewed all AI-assisted content and bear full responsibility for the final submission.

