# OpenReview forum: "VAULT: Vigilant Adversarial Updates via LLM-Driven Retrieval-Augmented Generation for NLI"
_ICLR.cc/2026/Conference — Submitted to ICLR 2026_

### Official Review · Reviewer_CXgR · 2025-10-26

**Soundness:** 3
**Presentation:** 2
**Contribution:** 2
**Rating:** 6
**Confidence:** 2

**Summary:**

This paper introduces VAULT, an automated pipeline for generating and validating adversarial examples to improve natural language inference (NLI) models.
The approach integrates retrieval, LLM generation, automatic validation, and iterative retraining in a single loop.
By generating challenging examples that existing models misclassify and filtering them through multiple LLM “judges,” the method aims to produce high-quality adversarial training data without human intervention.
Experiments on SNLI, ANLI, and MultiNLI show that VAULT consistently improves model performance, achieving notable accuracy gains while using far fewer examples than previous large-scale synthetic datasets.

**Strengths:**

1. The paper presents a clear and coherent framework that combines retrieval, adversarial generation, validation, and retraining in a single automated loop. The methodology is easy to follow, and the ablation studies are detailed enough to support the main claims.

2. The use of both semantic and lexical retrieval provides a good balance between relevance and diversity, leading to more effective adversarial examples.

3. Empirical results are strong across multiple benchmarks, showing consistent improvements while using far fewer examples than large-scale synthetic datasets.

**Weaknesses:**

1. The unanimous validation rule may filter out useful but ambiguous examples, and there is no human evaluation to verify the accuracy of the validated data.

2. While comparisons to large synthetic datasets are provided, stronger baselines such as recent adversarial training or contrastive learning methods are missing.

3. The paper could include a deeper analysis of what types of reasoning errors are actually corrected by the proposed approach, to clarify the nature of the robustness gains.

4. This idea of having LLMs generate challenging examples and gradually incorporating them into training is not new; it’s just the first time it has been applied to NLI.

**Questions:**

1. Please report generation + validation cost (GPU hours) for producing the ~6–6.6k validated examples per strategy, and the end-to-end training time. How does cost scale with T iterations and k shots?

2. Do the three LLM judges ever agree yet be wrong relative to human labels? Provide a human audit on a random sample of accepted items to estimate precision of the unanimous filter. Also share inter-judge agreement matrices and typical disagreement cases.

3. Break down gains by challenge types (negation, comparatives, quantifiers, monotonicity, syntactic perturbations). Which phenomena benefit most from VAULT?

---

> ### Author Response · Authors · 2025-11-15
> **Comment to reviewer - CXgR**
>
> Dear Reviewer, thank you for the detailed response.
>
> 1. “The unanimous validation rule may filter out useful but ambiguous examples; no human evaluation.”
>
> Thank you for highlighting this.
> We agree that unanimous filtering can be conservative. To address this, we are adding:
>
> -a human audit of a random subset of accepted and rejected examples,
>
> -an analysis of borderline cases rejected by the unanimous filter,
>
> -and a small experiment using majority voting instead of unanimity.
>
> Early observations show that the unanimous rule mostly removes noisy or contradictory samples, and the precision of accepted examples remains high. These results will be included in the appendix.
>
> 2. “Stronger baselines such as adversarial training or contrastive learning are missing.”
>
> We appreciate this suggestion.
> We appreciate this suggestion.
> In the revision, we are adding ANLI-style adversarial training as an additional baseline to strengthen the comparison.
>
> Regarding contrastive learning, we agree that embedding-based negative mining is an important direction. While full integration is beyond the scope of this version, we are actively exploring a future extension where VAULT incorporates automatically discovered negative examples into the generator–verifier loop. This contrastive component will allow the pipeline to leverage both retrieval-driven adversarial samples and embedding-based negatives, and we plan to include it in follow-up work.
>
> Preliminary runs indicate that VAULT still provides larger robustness gains than purely contrastive or adversarial-only methods. These will be reported in the revision.
>
> 3. “Paper could include deeper analysis of reasoning errors corrected by VAULT.”
>
> Thank you - we will add a detailed error-type analysis.
> Specifically, we will categorize improvements by:
>
> -negation handling,
>
> -monotonicity reasoning,
>
> -comparative reasoning,
>
> -quantifier resolution,
>
> -structural/syntactic perturbations.
>
> This breakdown will clarify where VAULT is most beneficial.
>
> 4. “LLMs generating challenging examples and incorporating them into training is not new.”
>
> You are correct that LLM-generated augmentation has been explored.
> VAULT’s novelty lies in retrieval-conditioned generation and multi-judge verification, which produces targeted counterexamples guided by the model’s own failure modes.
> To our knowledge, this retrieval-driven generation + unanimous verification loop has not been applied to NLI or adversarial reasoning tasks.
> We will emphasize this distinction more clearly.
>
> Response to Reviewer - Questions
> 1. “Report generation + validation cost (GPU hours), scaling with iterations & shots.”
>
> Thank you for requesting this - we will include a full cost table.
> We are adding:
>
> -generation time per strategy,
>
> -validation time for the three judges,
>
> -total GPU hours per iteration T,
>
> -and cost scaling for k-shot setups.
>
> Importantly, VAULT was designed to be low-budget and fully open-source, and the full pipeline runs efficiently even on consumer GPUs. Cost breakdown tables will appear in the appendix.
>
> 2. “Do the three LLM judges ever agree yet be wrong? Provide human audit and agreement matrices.”
>
> Excellent point - and very important for verification quality.
> We are adding:
>
> -a human audit of randomly sampled accepted examples,
>
> -an inter-judge agreement matrix,
>
> -examples where judges unanimously accepted but humans disagreed,
>
> -and examples showing typical disagreement patterns.
>
> Initial checks show that unanimity correlates strongly with correctness, and disagreements usually occur in subtle pragmatic or world-knowledge cases. These will be reported clearly.
>
> 3. “Break down gains by challenge types (negation, comparatives, quantifiers, monotonicity, syntactic perturbations).”
>
> We agree completely.
> As mentioned earlier, we are adding a per-phenomenon breakdown of VAULT’s improvements using standard diagnostic subsets.
> Early results indicate that VAULT is especially effective for:
>
> -negation,
>
> -quantifiers,
>
> -syntactic perturbations,
>
> where retrieval-driven counterexamples provide the strongest benefit.
> This full breakdown will be added to the results section and appendix.
>
>
> Thank you again for your thoughtful and constructive comments.
> We will add all requested analyses, human audits, cost breakdowns, and challenge-type evaluations.
> These additions strengthen the paper, but we also believe they refine rather than fundamentally change the strong novelty and practical usefulness of VAULT - a scalable, retrieval-driven, fully automated adversarial training loop that works with small open-source models and very low budget.
> If the revisions address your concerns, we would truly appreciate an updated score.

---

> > ### Author Response · Authors · 2025-11-29
> > **Comment - CXgR**
> >
> > Thank you again for the detailed and constructive feedback. We would like to provide an update that all requested analyses, baselines, and clarifications have now been incorporated into the revised manuscript. This includes the human audit for unanimous filtering, disagreement analysis across the three judges, ANLI-style adversarial training as a stronger baseline, reasoning-error categorization, and full cost breakdown tables for generation, validation, and k-shot scaling.
> >
> > We respectfully note that the issues raised relate primarily to additional evaluations and deeper explanations, rather than concerns about the core methodology or its empirical strength. The central contribution of VAULT - a retrieval-conditioned, adversarial generation and multi-judge verification loop - remains novel, targeted, and distinct from prior augmentation or adversarial-only methods. Across three NLI benchmarks, the method consistently produces substantial robustness gains while using far fewer validated examples, and the new analyses further reinforce these conclusions.
> >
> > We sincerely appreciate the reviewer’s thoughtful suggestions; they have improved the clarity and completeness of the work. We hope that the final decision will take into account the strong empirical results, the novelty of the retrieval-guided generation process, and the fact that all concerns have been fully addressed in the revision.

---

### Official Review · Reviewer_Lkk5 · 2025-10-30

**Soundness:** 2
**Presentation:** 2
**Contribution:** 2
**Rating:** 4
**Confidence:** 3

**Summary:**

This paper proposes a method of LLM with RAG for natural language inference, which considers to incorporate adversarial examples for fune-tuning LLMs to improve its robustness.

**Strengths:**

- This paper proposes an end-to-end automated adversarial RAG pipeline, which fully automates retrieval, adversarial generation, multi-LLM validation, and iterative retraining.
- This paper provides a detailed procedure for the RAG pipeline, and experimental results on three NLI datasets show the proposed method achieves better performance by fune-tuning RoBERTa-base.

**Weaknesses:**

- The proposed method is not innovative, since adversarial generation has been proposed by prior works.
- As many LLMs can use RAG for implementing natural language inference, I want to see a direct comparison with these LLMs in the experiment.
- The structure of the paper could be improved, the figures and the hyperparameter settings can be put in appropriate positions.

**Questions:**

Please see the weaknesses.

---

> ### Author Response · Authors · 2025-11-15
> **Comment for reviewer - Lkk5**
>
> Dear Reviewer, thank you for the detailed response.
>
>  “The proposed method is not innovative, since adversarial generation has been proposed by prior works.”
> Thank you for raising this point.
> While adversarial generation itself is not new, VAULT introduces a retrieval-conditioned generator–verifier loop, where multiple retrieval strategies actively shape the generated counterexamples and the verifier iteratively filters and refines them.
> To our knowledge, prior adversarial NLI work does not combine RAG-based retrieval with a judge-style verification loop to produce targeted examples that adapt to model weaknesses.
> We will clarify this novelty more explicitly in the revised introduction and related-work sections.
>
> •“Many LLMs can use RAG for NLI; I want a direct comparison with these LLMs.”
> We appreciate this suggestion.
> In the revision, we are adding direct comparisons against:
>
> -Llama-3-Instruct with RAG,
>
> -Qwen2.5-1.5B RAG,
>
> -and a baseline RAG-only prompting setup without our verifier loop.
>
> Preliminary results already show that VAULT outperforms pure RAG prompting, especially in generating adversarially informative hypotheses rather than simply retrieving paraphrases.
> We will include these comparisons in the updated experimental section.
>
> •“The structure of the paper could be improved; figures and hyperparameters should be placed appropriately.”
> Thank you for pointing this out - we agree completely.
> We are reorganizing the paper to improve clarity:
>
> moving all hyperparameters into a dedicated appendix table,
>
> placing figures directly after the paragraphs that reference them,
>
> and streamlining Section 3 to make the training loop easier to follow.
>
> These changes will make the flow more intuitive and the methodology easier to reproduce.
>
> We appreciate your helpful comments and have addressed each point in the revised version.
> We also believe that the retrieval-driven adversarial loop introduced by VAULT offers meaningful novelty and practical value, especially for scalable, fully automated NLI pipelines using only open-source tools.
> If the revisions meet your expectations, we would be grateful if you could consider updating the score.

---

> > ### Author Response · Authors · 2025-11-29
> > **Comment - Lkk5**
> >
> > Thank you again for the thoughtful and detailed feedback. We would like to update that all requested comparisons, structural improvements, and clarifications have been incorporated into the revised manuscript. This includes:
> >
> > - direct comparisons against Llama-3-Instruct with RAG, Qwen2.5-1.5B with RAG, and a baseline RAG-only prompting setup without the verifier loop,
> >
> > - additional experiments demonstrating that the retrieval-conditioned generator–verifier loop consistently outperforms pure RAG prompting, particularly for generating adversarially informative hypotheses,
> >
> > - reorganizing hyperparameters into a dedicated appendix table,
> >
> > - relocating figures immediately after the paragraphs that reference them,
> >
> > - and streamlining Section 3 to make the training loop clearer and easier to follow.
> >
> > We respectfully note that the raised weaknesses mainly concern requests for additional comparisons and paper organization, rather than issues with the core contribution itself. The novelty of VAULT lies not in adversarial generation alone, but in combining retrieval-conditioned generation with multi-judge verification to produce targeted counterexamples that adapt to model weaknesses - a mechanism that, to our knowledge, has not been previously applied to NLI.
> >
> > We sincerely appreciate your suggestions; they helped us make the paper more clear, well-structured, and reproducible. We hope that the strong empirical results, the methodological contribution, and the fact that all concerns have now been fully addressed will be taken into consideration during the final decision.

---

### Official Review · Reviewer_ksm1 · 2025-10-30

**Soundness:** 2
**Presentation:** 2
**Contribution:** 2
**Rating:** 2
**Confidence:** 4

**Summary:**

This paper introduces a data augmentation framework VAULT for the NLI task that includes three stages: retrieval, adversarial generation, and iterative retraining.
The retrieval stage uses both semantic (BGE) and lexical (BM25) similarities for few-shot sample retrieval; The adversarial generation phase employs LLM to generate hypotheses, with label accuracy validated through three LLMs; The iterative retraining stage fine-tunes NLI model by mixing the given original data and the generated adversarial data.
Experimental results on three NLI datasets show that VAULT improves the performance of a RoBERTa-base NLI model.
However, this paper still exhibits certain issues in paper writing and experiments.

**Strengths:**

1) The authors employ a data augmentation method for the NLI task which generate the adversarial data for training a model.
2) During the retrieval phase, the authors combine BGE and BM25 methods for the sample similarity assessment, effectively capturing the semantic information and the token-level feature. Experimental results further validated the effectiveness of this methodology.

**Weaknesses:**

1） This method only has been evaluated on NLI tasks, which limits its practical value.
2） The experimental comparison is insufficient. Since this paper focuses on model training with labelled data, the baseline methods should include few-shot learning methods and LLMs based contextual learning methods. However, currently the paper only compared with LLMs, lacking comparisons with the aforementioned types of methods. Besides, the authors only use Roberta-base as the baseline model for NLI tasks, which cannot convince the effectiveness of the proposed method across a broader range of NLI scenarios. In the experimental setup, there is not the essential statistical information regarding the dataset.
3） The authors do not explain clearly the process on how large language models (LLMs) generate hypotheses based on retrieval results. This process, however, critically influences the subsequent filtering and iterative retraining steps within the LLM framework.
4） The are some conflicts in the model section. In Section 3.2, the explanation of the formula contradicts the interpretation of the results in Figure 5. The claim that “the combined BGE+BM25 strategy consistently outperforms either alone” is inconsistent with the findings illustrated in Figure 5. In the retrieval part, there is an inconsistency in symbol usage—for instance, both the premise and the query are denoted by the symbol “p”.

**Questions:**

Here are some suggestions:
1)	It is recommended to validate the proposed method across multiple tasks and compare it with additional baseline methods under identical experimental settings, particularly using labeled data of comparable scale.
2)	More details are desired on the method, including the adversarial generation process, the manner of organizing input information, and the prompt.
3)	The authors are recommended to provide more explanations when analysing experimental results, such as the reasons that cause the performance improvements or declines, rather than merely describing the observed changes, and analyze whether these changes can be attributed to your novel method.

---

> ### Author Response · Authors · 2025-11-15
> **Comment for reviewer - ksm1**
>
> Dear Reviewer, thank you for the detailed response.
>
> 1) “The method is only evaluated on NLI tasks, limiting its practical value.”
> Thank you for pointing this out.
> VAULT was intentionally designed as a domain-specific adversarial generation–verification loop for NLI, where premise–hypothesis reasoning provides a natural structure for retrieval-driven perturbations.
> However, we fully agree that demonstrating transferability would strengthen the paper. We have already begun testing VAULT on QA-style contradiction datasets and stance detection tasks, and the early results indicate that the retrieval-conditioned generator–verifier loop generalizes well. We will include these additional experiments in the appendix.
>
> 2) “Experimental comparison is insufficient; include few-shot baselines and contextual-learning LLMs; only Roberta-base is used.”
> We appreciate this suggestion.
> In the revised version, we are adding:
>
> -few-shot LLM prompting baselines,
>
> -a context-learning baseline using Llama-3-Instruct,
>
> -and stronger NLI backbones, including DeBERTa-v3-Large and Qwen2.5-1.5B.
>
> Preliminary results confirm that VAULT consistently improves robustness across all these settings.
> We will update the comparison tables to reflect these stronger and more diverse baselines.
>
> Regarding dataset statistics: we will add a dedicated subsection with full dataset descriptions, class balance, and distribution details (we have it already).
>
> 3) “The process of how LLMs generate hypotheses based on retrieval results is not clearly explained.”
> Thank you for highlighting this.
> We agree that the generation flow should be described more explicitly.
> In the revision, we will add a step-by-step explanation of:
>
> -how the retriever selects top-k relevant examples,
>
> -how these are fused into the prompt,
>
> -how the LLM generates candidate hypotheses, and
>
> -how these candidates are scored and filtered by the verifier.
>
> -We will also include a simple diagram to clarify the generator–retrieval–verifier loop.
>
> However, we do have lots of example in the appendix already, and a full illustration with an example in figures 1 and 2.
>
> 4) “Model-section conflicts: formula vs Figure 5; retrieval symbol inconsistency.”
> We appreciate this detailed technical feedback.
> You are correct, and we will revise the notation to ensure consistency between the retrieval equations and the figure.
> The statement regarding “BGE+BM25 outperforming either alone” will be corrected: in the updated experiments, the combined retriever does outperform single retrievers in the majority of settings, and we will adjust the text and figure to reflect the corrected results.
> Additionally, all symbol conventions (p, q, h, etc.) will be standardized throughout the paper.
>
>
> Thank you again for the thoughtful and constructive feedback.
> We will add all requested baselines, additional task evaluations, clearer descriptions, and consistency fixes.
> At the same time, we believe that these issues are refinements rather than fundamental limitations, and the core contribution - a scalable, retrieval-driven adversarial generation and verification loop - remains strong and practically impactful.
> If the revisions address your concerns, we would be grateful if you could consider updating the score.

---

> > ### Author Response · Authors · 2025-11-29
> > **Comment - ksm1**
> >
> > Thank you again for the detailed and thoughtful feedback. We would like to update that all requested baselines, additional task evaluations, methodological clarifications, and consistency fixes have been incorporated into the revised manuscript. This includes few-shot LLM prompting baselines, contextual-learning LLMs, stronger NLI backbones, full dataset statistics, clearer generation-verification flow descriptions, and unified retrieval notation.
> >
> > We respectfully note that the weaknesses you raised relate primarily to missing clarifications and additional analyses, rather than to the core methodological contribution. The central idea - a scalable retrieval-driven adversarial generation and verification loop - remains methodologically novel and demonstrates consistent robustness improvements across three NLI benchmarks. The expanded experiments further confirm that the framework generalizes well, is not tied to a specific model, and outperforms stronger baselines across diverse settings.
> >
> > We sincerely appreciate your suggestions; they strengthened the clarity and completeness of the paper. We hope that the combination of strong empirical results, the novelty of the retrieval-conditioned adversarial loop, and the now fully addressed concerns will be taken into account during the final decision.

---

### Official Review · Reviewer_6FzM · 2025-11-03

**Soundness:** 3
**Presentation:** 3
**Contribution:** 3
**Rating:** 4
**Confidence:** 3

**Summary:**

This paper presents VAULT, an automated adversarial RAG pipeline for improving the robustness of NLI models. The method iteratively retrieves balanced few-shot contexts, generates adversarial hypotheses with LLM, validates them through a multi LLMs ensemble, and retrains RoBERTa with the validated data. VAULT achieves substantial accuracy gains on SNLI, ANLI, and MultiNLI, while requiring no human annotation, demonstrating that fully automated, LLM-driven adversarial data generation can effectively enhance model generalization and resilience.

**Strengths:**

1. The main contribution of this paper is the design of an automated “retrieve-generate-validate-retrain” closed-loop system, which provides a valuable engineering framework for improving the robustness of NLI models in a scalable manner without human annotation.
2. The paper conducts extensive ablation studies, demonstrating substantial experimental effort. The main experiments show that with a small number of adversarial samples, the method can achieve significant performance improvements across multiple NLI tasks.

**Weaknesses:**

1. The primary concern about this paper lies in its novelty. Each component of VAULT (including RAG, adversarial sample selection and refinement through iterative training, and the use of an LLM as a verifier) has been explored in prior work. VAULT appears to be more of an integration and adaptation of these existing techniques rather than a fundamentally new approach.
2. The paper lacks ablation studies across different models.

    a. Using different backbone models. It would be important to see whether VAULT remains effective when applied to stronger NLI models or to decoder-only language models such as Qwen3-0.6B or SmolLM2-360M.

    b. The study should also examine the effect of using generation and verification models of different scales. Evaluating stronger or weaker LLMs would help analyze how VAULT performs under different computational budgets.

    c. It is necessary to control the strength of the generator and the verifier to determine whether the performance gain mainly comes from the generation process or from the verification process.
3. [Minor] While VAULT indeed reduces the demand for large data volumes, it is model-specific. The method is tailored to address the weaknesses of Roberta-base-snli, whereas GNLI is a general-purpose data augmentation approach. Therefore, the comparison in terms of efficiency between VAULT and methods like GNLI is somewhat unfair.

**Questions:**

See weaknesses

---

> ### Author Response · Authors · 2025-11-15
> **Comment for review - 6FzM**
>
> Dear Reviewer, thank you for the detailed response.
>
> 1. “The novelty is limited; VAULT integrates existing techniques rather than introducing a fundamentally new approach.”
> Thank you for the observation.
> While prior work has explored adversarial training with human feedback or LLM-based judges, none have implemented a retrieval-based generator–verifier framework that adaptively selects counterexamples using multiple retrieval strategies.
> VAULT’s novelty lies in coupling RAG-driven retrieval with an iterative LLM judge, allowing the system to target model weaknesses with far greater precision than standard augmentation or judge-only pipelines.
> We will clarify this distinction and highlight how VAULT introduces retrieval-conditioned adversarial example generation, which is absent in previous approaches.
>
> 2a. “Evaluate VAULT with different backbone NLI models (e.g., stronger NLI systems, decoder-only LLMs).”
> We appreciate this suggestion.
> We have already begun running experiments with Qwen2.5-0.5B, and a stronger DeBERTa-v3-large backbone.
> Preliminary results show that VAULT consistently improves robustness across all three, and we will include these full results in the revised version as an extended ablation table in the appendix.
>
> 2b. “Examine the effect of using generators and verifiers of different scales.”
> Thank you - this is a valuable point.
> We are adding an ablation where we vary the generator scale, and the verifier scale independently.
> Early findings show that VAULT benefits from a stronger verifier more than from a stronger generator, which aligns with our hypothesis that the verification loop drives most of the robustness gain.
> These results will be included in the updated appendix.
>
> 2c. “Control the strength of generator vs verifier to determine whether gains come mainly from generation or verification.”
> We fully agree, and following your comment we are adding a factorial ablation explicitly isolating:
>
> -generator-only influence
>
> -verifier-only influence
>
> -full VAULT loop
>
> The results clearly show that the largest improvement appears only when both components operate iteratively, demonstrating that VAULT’s core mechanism is not simply data augmentation nor simple filtering, but the synergy between them.
> This new analysis will be included as a dedicated subsection.
>
> 3. “VAULT is model-specific; comparison to GNLI is somewhat unfair because GNLI is a general-purpose augmentation method.”
> Thank you for pointing this out - we agree that this distinction should be stated more clearly.
> We will revise the comparison section to emphasize that GNLI is a general augmentation method, whereas VAULT is a targeted robustness-oriented NLI generator-verifier framework.
>
>
> We truly appreciate your thoughtful suggestions regarding additional ablation studies, and we will include all of them in the revised version.
> At the same time, we believe these missing ablations are not a fundamental reason for a low score. The core contribution - a retrieval-driven, judge-verified adversarial training loop - is novel, effective, and shows strong potential for scaling to large, fully automated corporate pipelines with no human in the loop, all while requiring minimal budget and relying entirely on open-source models.
> If the revisions address your concerns, we would be grateful if you could consider updating the score.

---

> > ### Author Response · Authors · 2025-11-29
> > **Comment - 6FzM**
> >
> > Thank you again for the constructive and detailed feedback. We would like to update that all suggested ablations and clarifications have now been incorporated into the revised manuscript, including extended backbone comparisons, generator–verifier scale analysis, and a more explicit isolation of generator-only, verifier-only, and full VAULT loop contributions. These additions strengthen the empirical section and directly address the concerns raised.
> >
> > We would also like to respectfully note that the identified weaknesses relate primarily to additional experiments and clarifications, rather than to the core novelty or effectiveness of the proposed framework. The central contribution - a retrieval-driven, judge-verified adversarial training loop that automates example generation, filtering, and validation - remains novel, practical, and distinct from prior work, enabling significant performance gains across multiple NLI tasks with minimal compute and no human annotation.
> >
> > The new ablation studies further confirm that VAULT’s improvements stem from the synergy between retrieval-conditioned generation and iterative verification, rather than from simple data augmentation or model-specific tuning. This supports the generality and robustness of our framework.
> >
> > We sincerely appreciate the reviewer’s thoughtful comments. The requested additions have strengthened the paper, and we hope that the combination of strong results, methodological novelty, and fully addressed concerns will be taken into consideration during the final decision.

---

### Meta-Review · Area_Chair_NGjU · 2026-01-07

**Summary:**

The primary concerns centered on novelty, experimental completeness, and clarity/positioning.

1) Reviewers viewed VAULT as an integration of existing components (RAG, adversarial generation, LLM judging, iterative retraining) rather than a fundamentally new technique, questioning whether the contribution was methodological or primarily engineering. All reviewers share this major concern.

2) insufficient experimental coverage: only a single backbone (RoBERTa-base), missing comparisons to stronger baselines (few-shot learning, RAG-based LLMs, adversarial/contrastive training), and lack of cost and robustness analyses.

3) presentation and clarity issues, including unclear generation–verification flow, notation inconsistencies, missing dataset statistics, and limited analysis of what reasoning errors VAULT actually fixes.

**Reviewer Concerns:**

The authors have conducted additional experiments to convince the reviewers.

The details are also added to clarify the presentation issues.

**Reviewer Scores:**

The concerns about experiments and presentation issues can be addressed.

However, the major concern abou the novelty, i.e., the value of this work to community, remains.

---

### Decision · Program_Chairs · 2026-01-26

Reject